# In-memory mechanical computing

Tie Mei[1] & Chang Qing Chen [1] ✉

Mechanical computing requires matter to adapt behavior according to retained knowledge, often through integrated sensing, actuation, and control of deformation. However, inefficient access to mechanical memory and signal propagation limit mechanical computing modules. To overcome this, we developed an in-memory mechanical computing architecture where computing occurs within the interaction network of mechanical memory units. Interactions embedded within data read-write interfaces provided function-complete and neuromorphic computing while reducing data traffic and simplifying data exchange. A reprogrammable mechanical binary neural network and a mechanical self-learning perceptron were demonstrated experimentally in 3D printed mechanical computers, as were all 16 logic gates and truth-table entries that are possible with two inputs and one output. The in-memory mechanical computing architecture enables the design and fabrication of intelligent mechanical systems.

Mechanical computing is an unconventional information processing method where information is stored as deformation states and computation is conducted via mechanism motion or deformation evolution[1,2]. Such mechanical computing is desirable for intelligent mechanical systems[2–5], including applications such as soft devices[6], micro-electro-mechanical systems[7], and robotic materials[8,9]. Here, intelligence represents an ability to adapt behavior according to retained knowledge[10,11], and intelligent matter thus requires a memory module to store data, a computing module to process data and adapt the matter, and a data exchange strategy for communication between the memory and computing modules. The feasibility of such a mechanical system was established by Babbage in the 19th century and has been demonstrated many times since[2–5,12–22]. Although mechanical computing offers unique potential associated with data security and resistance to electromagnetic interference[21–23], it is limited in performance compared to silicon-based electronic computing, with the data exchange strategy being a key limiting factor. Our goal was therefore to design a mechanical computing architecture that circumvents this limitation.

The fundamental units of computing modules, logic gates, have been explored in mechanical computing in the form of origami[12,13], buckled beams[14], and linkages[22]. Signal propagation can occur through these logic gates via mechanical[3,22], mechano-electronic[4,5], or mechanical-fluidic[15,16] interfaces, and a reprogrammable, von Neumann-like mechanical computing architecture has been demonstrated[17]. With advances in fabrication techniques and materials[18], mechanical computing systems can be constructed on different scales and can process environmental information such as pressure[19] and chemical cues[20].

Similarly, non-volatile mechanical memory modules exist for data storage[24,25], based on bi-stable elements[26–29]. However, the reading and writing of data in these modules requires either the use of complex peripheral equipment[27] (e.g., movable electromagnetic coils) or the evolution of the whole system across intricate state transition paths[26,28]. These are effective, but require the design of elaborate energy landscapes to control energy flow through the mechanical system[30–32], and are the source of a critical bottleneck that limits the bandwidth of mechanical signal propagation between the memory and computing modules.

To address this bottleneck, in-memory computing has been explored[33–35], an information processing framework wherein, similar to the human brain[36], computation occurs in the places where data are stored. Integrating computing and memory modules within a system framework provides for widely distributed interfaces between data and computing. This has been effective in semiconductor non-volatile memory devices[37], with in-memory computing especially efficient for data-centric and intelligent tasks[38]. Our rationale for exploring this in the context of mechanical computing is that in-memory computing reduces the "distance" between computing and data, and potentially the associated data traffic bottleneck as well.

Here, we propose an in-memory mechanical computing architecture, aiming at satisfying the requirement of being framed by

---

[1]Department of Engineering Mechanics, CNMM and AML, Tsinghua University, Beijing 100084, PR China. ✉e-mail: chencq@tsinghua.edu.cn

distributed, non-volatile, robust, readable, initializable, and reprogrammable mechanical memory units, similar to its electronic counterpart[35]. Additionally, the interactions between memory units should form function-complete logic sets to ensure the realization of a variety of algorithms, and these interactions should be compatible with the basic neuromorphic computing models to facilitate artificial intelligence. To meet the three requirements, our architecture uses a network of non-volatile binary mechanical memory units (buckled beams) that can be reprogrammed and initialized by an external force. Computing in this architecture is driven by a time-varying signal (a periodic external force) and determined by the network topology with three basic interactions: a shift register, an XNOR gate, and a perceptron operation. Experimental demonstrations of this architecture to develop reprogrammable mechanical binary neural networks and a self-learning mechanical perceptron indicate that the proposed architecture satisfies all three requirements.

## Results

### General concept of in-memory mechanical computing

The proposed in-memory mechanical computing architecture consists of interconnected square blocks, each representing a binary (i.e., $x = 0$ or 1) mechanical memory unit (Fig. 1a). The blocks have three basic interaction operations: a shift register (black arrows), an XNOR gate (curved arrows and plus the "$\odot$" operator), and perceptron operations (arrows with weight parameters $\alpha_j$). The state of the system at clock phase $t$ is described by all memory units (i.e., $x_i^{(t)}$, where the subscript $i$ refers to the $i$-th unit). Driven by a time signal (Fig. 1b), the system evolves to another state $(x_1^{(t+1)}, x_2^{(t+1)}, \ldots, x_n^{(t+1)})$ as simultaneous data writing, data reading, and computation occur in the mechanical memory unit array according to the organization of interaction networks.

The system evolutions give rise to computing processes for the three basic interaction operations that were our focus (Fig. 1c–e). The first was the shift register, a sequential logic device in electronic computers[39] that can store binary numerical code, perform serial-to-parallel data conversion, and provide a strategy for reading and writing serial data. In the proposed mechanical shift register (Fig. 1c), the state of an upstream memory unit (i.e., $x_{i-1}^{(t)}$) is transferred to the memory unit that is immediately downstream (i.e., $x_i^{(t+1)}$) after these units receive a time signal. With this operation, data with different clock phases can be written into the mechanical memory units in a tandem array to participate in computation together. Thus, the historical data influence subsequent decision-making in the mechanical system, which is a foundation of learning[10,11].

For the XNOR operation (Fig. 1d), the state of the output unit ($x_{i+2}^{(t+1)}$) is updated upon receipt of a time signal, in accordance with the logic of the XNOR gate (right inset, Fig. 1d) and the two input units $x_i^{(t)}$ and $x_{i+1}^{(t)}$. If the two inputs are the same, the output will be 1, and if they are different, it will be 0. The XNOR operation differentiates two binary data, which is beneficial for error analyses and for adaptation of the mechanical computing system.

For the perceptron operation (Fig. 1e), computation upon receipt of a time signal begins with a weighted summation of the binary inputs ($x_1^{(t)}$ to $x_n^{(t)}$), $\sum_{i=1}^{n} \alpha_i x_i^{(t)}$ where $\alpha_i$ are weights. The output ($x_{n+1}^{(t+1)}$) follows a nonlinear step function activation, with $x_{n+1}^{(t+1)} = 1$ if $\sum_{i=1}^{n} \alpha_i x_i^{(t)} \geq 1$, and 0 otherwise. The proposed mechanical perceptron is a variant version of the McCulloch-Pitts artificial neuron model[40], enabling a mechanical system with embedded neuromorphic arithmetic.

Note that this comprises a function-complete set of logic gates because the required three logic gates (AND, OR, and XNOR) can be achieved with these units. The AND logic gate is obtained from the mechanical perceptron when only two input units are included and $\alpha_i$ are set to 1/2. The OR gate is similarly obtained for two inputs and $\alpha_i$ set to 1. This function-complete set provides a vast functional design space with the proposed architecture.

The mechanical memory unit selected was a simple and well-studied bi-stable mechanical component, i.e., the buckled beam[41] whose state is distinguished by its arching direction (0 for arching to the left, and 1 for arching to the right), to benefit the experimental realization of the computing system. The design of mechanical

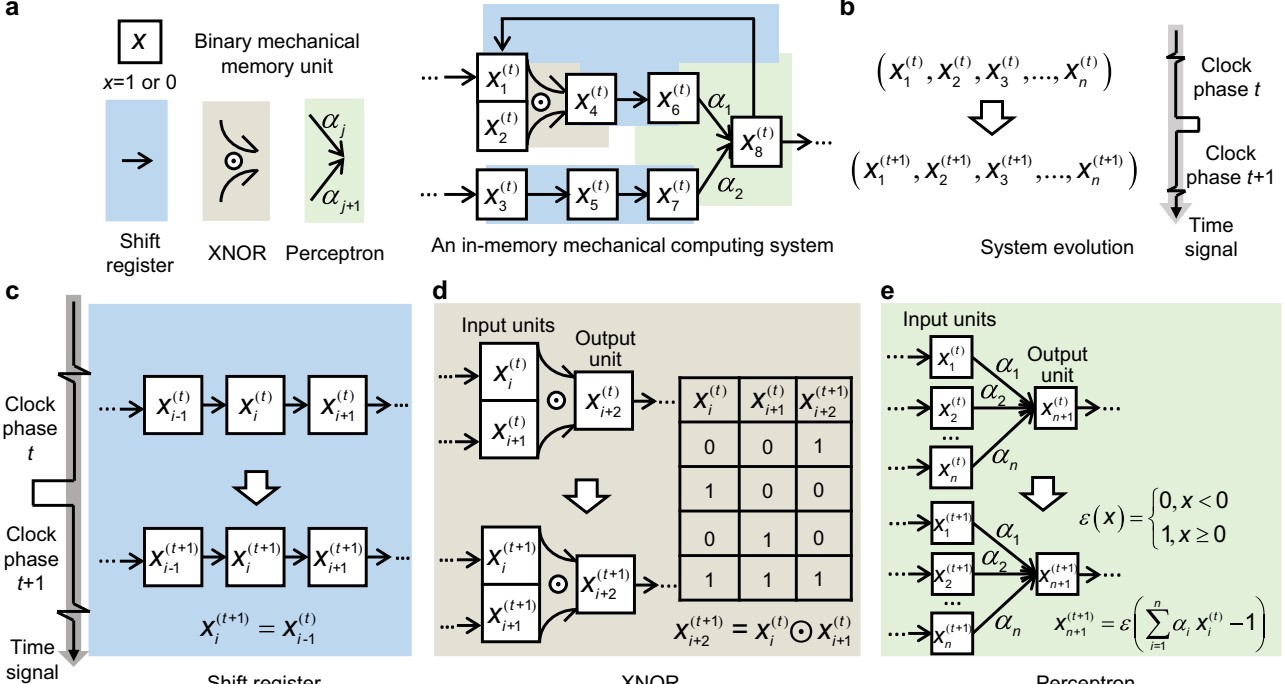

**Fig. 1 | Schematic of an in-memory mechanical computing system. a** An in-memory mechanical computing system consisting of binary mechanical memory units and their interactions (i.e., shifter register, XNOR, and perceptron operations). **b** Its computing process as the state evolution of the memory units. **c** Interaction serving as a shifter register. **d** Interaction serving as an XNOR gate, together with its truth table. **e** Interaction serving as a perceptron operation.

structures for the basic operations is given in the next section as a proof-of-concept for in-memory mechanical computing. And their underlying mechanism is discussed in the Method. These structures were fabricated by 3D printing, and the time signal (the external force $F$) was realized by a series of magnets in the experiments.

## Design of basic interactions

First, we introduce a mechanical structure for the shift register (Fig. 2). It adopts an up-down symmetrical design scheme and consists of links, springs, and a buckled beam, with their boundary conditions provided by the gray support also marked (Fig. 2a(i)). Note that the white blocks denote hollow space introduced to reduce the cost of 3D printing. The links are hinged together. Spring 1 is placed between the buckled beam and link 3, while two spring 2 are connected to the support and the common vertex of link 1. All springs can only provide a compressive interaction. The top and bottom ends of the buckled beam are fixed, and the midpoint of the beam can only translate horizontally. More details of assembling these components are given in the Supplementary Note. Figure 2b shows the theoretical compressive force-displacement curve ($f - u$) of the buckled beam where a bi-stable mechanism can be found and $u_M$ represents the unstable equilibrium state of the beam (details of the corresponding Euler buckling based mechanics are given in the Method). When the buckled beam arches to

the left (right), it is in state 0 (1). The bi-stable buckled beam can be used to store binary data and serves as a mechanical memory unit. When the beam is compressed to state 1, the links can move right almost unimpededly by considering spring 2 is much softer than spring 1, Fig. 2a(ii).

Operation of the shift register is driven by a time signal, i.e., the external force $F$ applied to the links shown in Fig. 2c. If the initial state of the mechanical memory unit is 0 before receiving $F$, the motion of link 1 will be blocked by the support, and thus all components cannot move, Fig. 2c(i). In addition, if the initial state of the mechanical memory unit is 1, the rhombus link 3 will be opened and push the left-buckled beam to state 0 via spring 1. Besides, all link 1 are not blocked until compressing the downstream unit for a distance of $\Delta$, Fig. 2c(ii). Then, as Fig. 2e shows, the buckled beam with a received displacement load $\Delta$ will arch to the right, regardless of its initial state of being 0 (Fig. 2e(i)) or 1 (Fig. 2e(ii)). The theoretical force-displacement curve of a buckled beam with the time signal will become mono-stable (Fig. 2d) because the right ends of spring 1 are fixed by the links blocked by the external force $F$. Figure 2d also shows the force condition of the buckled beams shown in Fig. 2c, e.

After the time signal is received and released by the memory units, one operation of the mechanical shifter register is completed, with the corresponding state evolution of the memory units

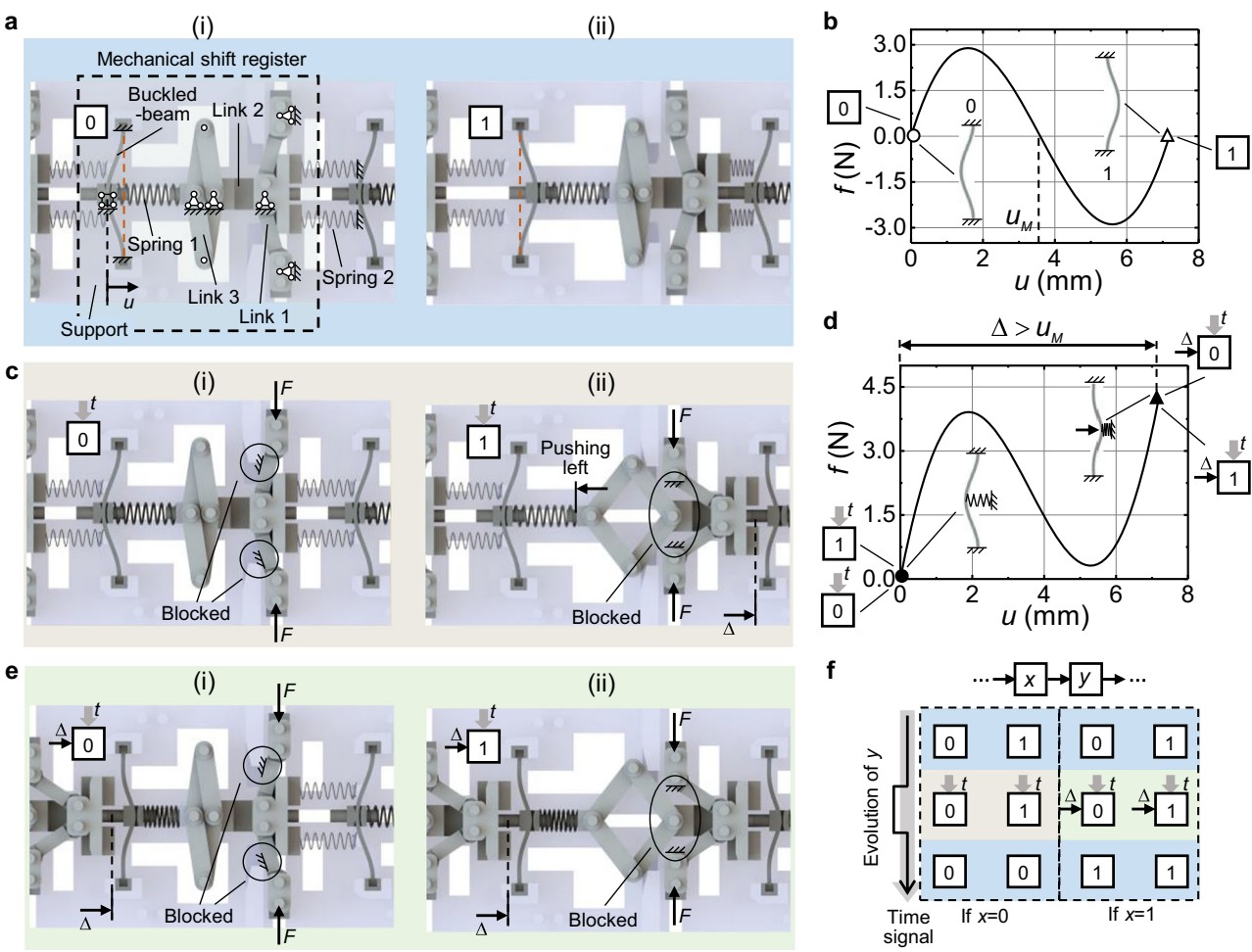

**Fig. 2 | Mechanical shift register. a** Structural design for mechanical shift register operation, comprising a binary mechanical memory unit (bi-stable buckled beam), links, springs, and support. The boundary condition of the components is also marked. The memory unit in (i) and (ii) is in state 0 and 1, respectively. **b** The theoretical compressive force-displacement response of the buckled beam in **a**. **c** Mechanical register structure receiving the time signal defined as the force $F$. For

(i) and (ii), the memory unit is at state 0 and 1 before being excited by $F$, respectively. **d** The theoretical compressive force-displacement response of the buckled beam in **c** and **e**. **e** Mechanical register structure receiving the time signal and the displacement load $\Delta$. For (i) and (ii), the memory unit is at state 0 and 1 before being excited, respectively. **f** State transformation map of the mechanical register unit.

given in Fig. 2f. Specifically, if the memory unit $x$ is in state 0, unit $y$ that receives a time signal does not get the displacement load $\Delta$. At this time, the buckled beam of unit $y$ will arch to the left, independent of its initial state (see Fig. 2c) because the beam is monostable (Fig. 2d). Then, after the time signal is released, the state of unit $y$ is 0. If the memory unit $x$ is in state 1, the buckled beam of unit $y$ with the time signal will be compressed to a distance of $\Delta$ and arch to the right (Fig. 2e). Then, unit $y$ becomes state 1 after releasing the time signal because $\Delta > u_M$. The state transmission of unit $x$ to unit $y$ showcases the shifter register operation.

Construction of the rest two mechanical operations, i.e., XNOR and perceptron is presented in Fig. 3. The mechanical structure of the XNOR operation has an up-down symmetry and consists of mechanical memory units, springs, balance bars, links, slider bar, slider block and support (Fig. 3a). The boundary conditions of these components are also marked. Details of the assembly of these components can be found in the Supplementary Note.

For XNOR, two memory units serve as the input ports and one serves as the output port. If the two inputs are 0 (Fig. 3a), there is no compressive stress in spring 1. The slider bars do not move. All balanced bars remain vertical, and the links stay at their initial position shown in Fig. 3a. Note that all link 1 point to the right under the condition. When imposed upon by a time signal (external force $F$), the output buckled beam is pushed for a distance of $\Delta$ (Fig. 3b). Thus, the output buckled beam arches to the right and its state switches to 1 after the force $F$ is released, i.e., the structure outputs 1 for the input (0, 0). On the other hand, if the input is (1, 0) (Fig. 3c), the buckled beam in state 1 compresses spring 1, thus pushing the slider bar, and the balance bars rotate. Then, driven by link 2 and link 3, the common vertex of link 1 moves left. After receiving the time signal, the vertex of link 1 further moves left for a distance of $\Delta$ (Fig. 3d). The slider block touches the slider bars, thus pushing spring 1 and initializing the input buckled beams to state 0. In this procedure, the output buckled beam is not loaded and always arches to the left. Thus, after releasing the time signal, the output is 0. Similarly, the structure can also output 0 when the input is (0, 1), and output 1 when the input is (1, 1). As such, the input-output relationship of the mechanical structure indeed matches the truth table of the XNOR gate in Fig. 1d. More details of the corresponding computing process of the structure are shown in the Supplementary Note.

For the structural design of the mechanical perceptron operation (Fig. 3e, f), two parallel mechanical shift register structures are connected to one output buckled beam via connecting bars and springs. The stiffness of the springs are $k_1$ and $k_2$, respectively. A critical stiffness of the spring, denoted as $k^*$, is defined in Fig. 3g. $k^*$ determines whether a buckled beam with the displacement load $\Delta$ will be compressed to state 1 and is used to evaluate the stiffness of springs in Fig. 3e, f, i.e., $k_i = \alpha_i k^*$, $i = 1$ or 2. Note that the input buckled beam 1 (2) is in state 1 (0) before receiving the time signal in Fig. 3e. Thus, the connecting spring 1 (2) will (not) compress the output buckled beam driven by force $F$. Considering $1/2 < \alpha_1 < 1$, the output buckled beam still arches to the left in Fig. 3e and will be state 0 after releasing the time signal. However, in Fig. 3f, the two input buckled beams are all in state 1 before receiving the time signal. All the parallel connecting springs compress the output buckled beam. This is equivalent to the condition thereby only one connecting spring of stiffness $(\alpha_1 + \alpha_2)k^*$ $(\alpha_1 + \alpha_2 > 1)$ compresses the output beam. As a result, the beam arches to the right, giving rise to state 1. Accordingly, the input-output relationship can be expressed as:

$$x_{n+1}^{(t+1)} = \varepsilon \left( \sum_{i=1}^{n} \alpha_i x_i^{(t)} - 1 \right) \tag{1}$$

with $n = 2$. In Eq. (1), $x_i^{(t)}$ represents the state of the input buckled beams before receiving the time signal, $x_{n+1}^{(t+1)}$ is the state of the output

buckled beam after the time signal is released, $\alpha_i$ is the dimensionless stiffness of the connecting spring, the summation originates from the parallel arrangement of the connecting springs, and the nonlinear activation stems from the binary state transformation of the output buckled beam. Moreover, $\sum_{i=1}^{n} \alpha_i x_i^{(t)}$ represents the interaction strength on the output buckled beam $x_{n+1}^{(t+1)}$. Note that the idea given in Fig. 3e, f can be applied to Eq. (1) with different $n$ by simply adopting more parallel springs. Thus, the mechanical perceptron operation in Fig. 1e is realized.

Two applications of the proposed in-memory computing architecture to intelligent mechanical systems are discussed in the following: a mechanical binary neural network and a mechanical self-learning perceptron.

## Mechanical binary neural network

Binary neural networks (BNN)[42,43] can reduce memory usage by training deep neural networks (DNN) with binary weights and activations and by replacing most multiplications with 1-bit XNOR operations. They are especially promising for deploying deep models on resource-limited devices such as mechanical computing systems. Here, a mechanical binary neural network (MBNN) is experimentally demonstrated with the help of the proposed in-memory computing architecture (Fig. 4).

In the binary neuron model of ref. 43 (Fig. 4a), the forward propagation procedure is based on the assumption of bipolar binary parameters, i.e., $\bar{x}_j^{l+1} = sign(\sum_{i=0}^{n} \bar{w}_{ij}^l \odot \bar{x}_i^l)$, where $\bar{x}_i^l$ and $\bar{x}_j^{l+1}$ are the input and output of the binary neuron, $\bar{w}_{ij}^l$ is the weight, superscript $l$ represents a layer rather than a clock phase to describe the networks more clearly, $\bar{x}_i^l, \bar{x}_j^l, \bar{w}_{ij}^l \in \bar{B}$, with $\bar{B}$ the set of bipolar binaries, i.e., ±1, and $\odot$ is the XNOR operation that outputs 1 (−1) when the two inputs are the same (different).

In the mechanical counterpart of the binary neuron model (Fig. 4b) the input-output procedure driven by the time signal can be written as $x_j^{l+1} = \varepsilon(\alpha \sum_{i=0}^{n} w_{ij}^l \odot x_i^l - 1)$ where $x_i^l, x_j^{l+1}, w_{ij}^l \in B$ (B is a set of binaries, i.e., 0 and 1), $\odot$ is the XNOR operation shown in Fig. 1d, $\varepsilon(x) = 0$ for $x < 0$, and $\varepsilon(x) = 1$ for $x \geq 0$. All connecting springs have the same stiffness, denoted by $\alpha$. This neuron model first computes the mechanical XNOR results of the input-weight pairs $(x_i^l, w_{ij}^l)$ and then outputs $x_j^{l+1}$ by performing a mechanical perceptron operation of these XNOR results.

It should be pointed out that, by properly setting $\alpha$, these two neuron models are equivalent (see Method). By way of example, if the parameter's value in the mechanical model (1 or 0) refers to the same information as that in the non-mechanical model (1 or −1), these two models have exactly the same function. Figure 4c shows a BNN model. By replacing the binary neuron with the corresponding mechanical one, an equivalent MBNN can be obtained (Fig. 4d), where $b_j^l$ and $\bar{b}_j^l$ are the bias of the $j$th neuron in the $l$th layer for the mechanical and non-mechanical neural networks, respectively. As explained in Methods, the dimensionless stiffness of the connecting springs ($\alpha$) for the first and second layers are set to be 1/3 and 1/2, respectively.

The MBNN can be made to execute the same function as the trained BNN. This is done by ensuring that the weights $w_{ij}^l$ and biases $b_j^l$ of the MBNN are chosen so that mechanical memory units yield 0 or 1 when the corresponding trained BNN would yield −1 or 1. We have trained a BNN that can judge the parity of input Morse code numbers 0–9 (the training arithmetic is discussed in the Supplementary Note). The corresponding MBNN in the experiment is given in Fig. 4e, where the values of the weight and bias are marked. The functions of several typical parts are also marked. The computing process of the MBNN (see Supplementary Movie 1) works in an asynchronous mode (Fig. 4f). First, a Morse code number enters the memory units of the first layer. Then, the state of the memory units in the second and third layers is computed, driven by two successive time signals. If the input number

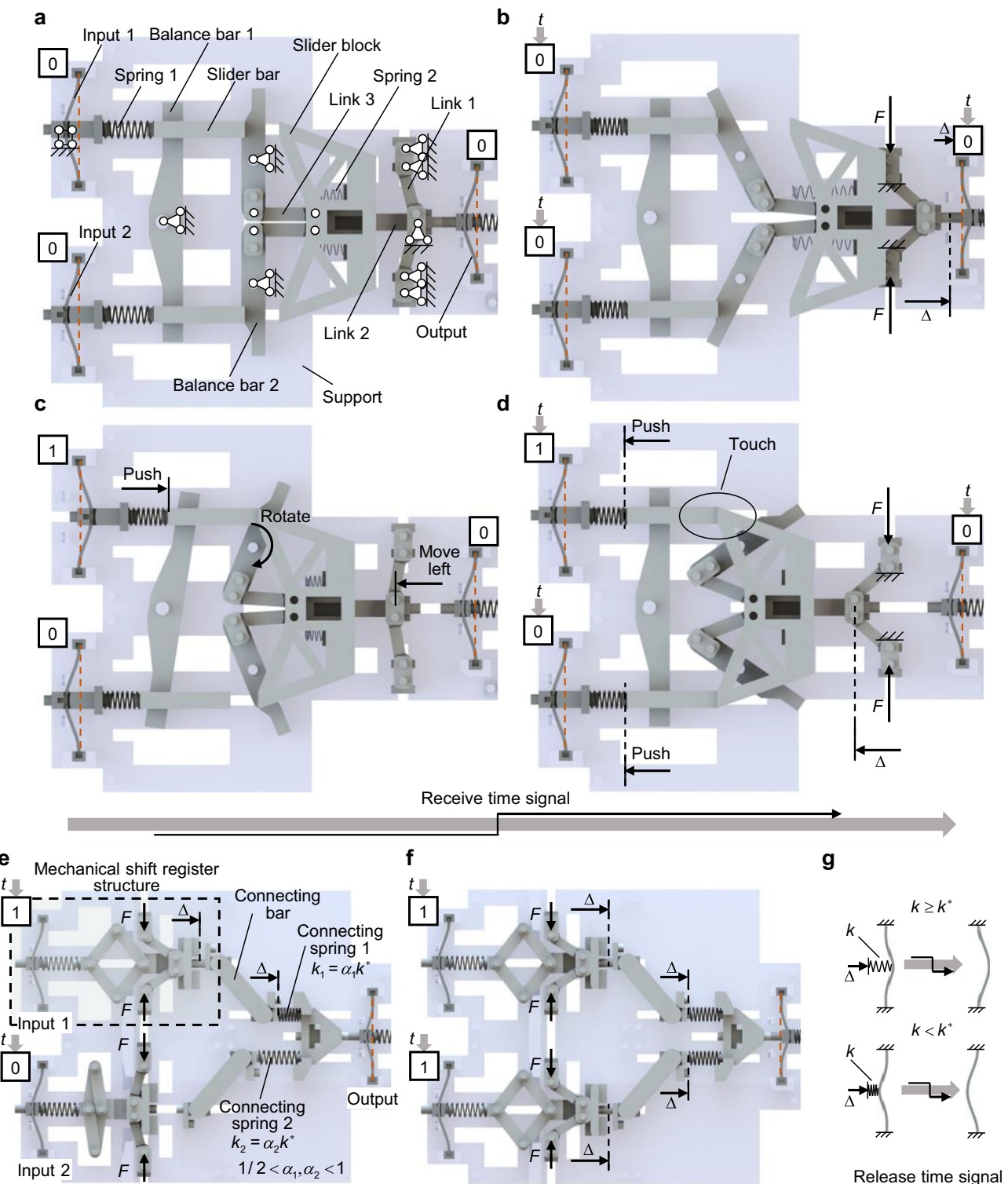

**Fig. 3 | Mechanical XNOR and perceptron operations. a** Structural design for mechanical XNOR operation before receiving a time signal. It consists of mechanical memory units, springs, balanced bars, links, slider bar, slider block, and support. The state of the left two memory units is (0, 0) and serves as input. **b** The mechanical XNOR structure in **a** after receiving a time signal. The right memory unit serves as output and will be state 1. **c** The mechanical XNOR structure with input (1, 0) before receiving a time signal. **d** The mechanical XNOR structure in **c** after receiving a time signal. It will output 0. **e** and **f** Construction of the mechanical perceptron operation. The input is the state of the left memory units before receiving the time signal, (1,0) in **e** and (1,1) in **f**. **g** Definition of the critical stiffness $k^*$ which determines whether the compressed buckled beam will become state 1 after releasing the time signal. If $k > (<) k^*$, the compressed beam arches to the right (left) and becomes state 1 (0) after releasing the time signal.

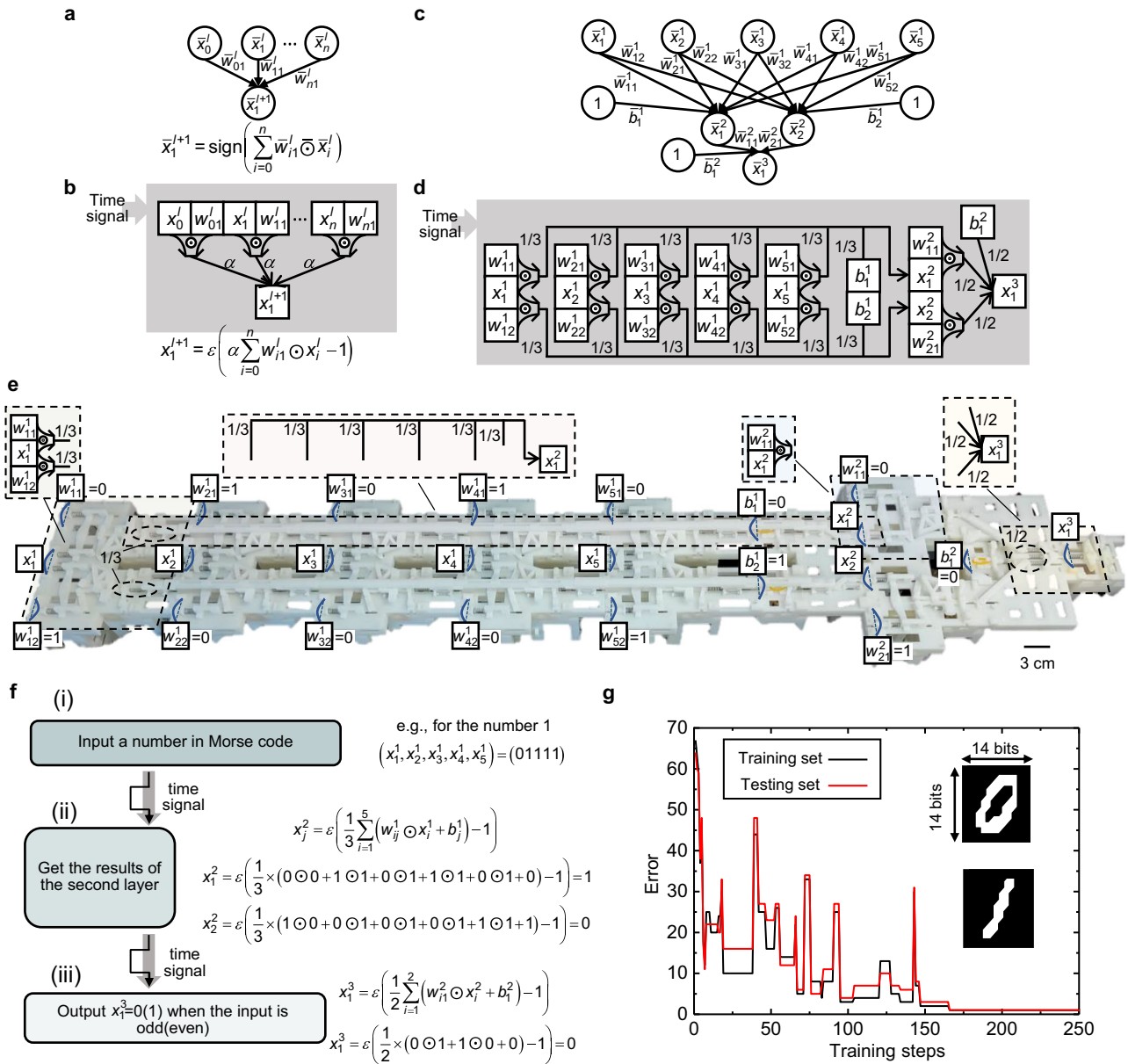

**Fig. 4 | Mechanical binary neural network. a** Binary neuron model. **b** An equivalent mechanical binary neural model corresponding to **a**. Once the system receives a time signal, it executes the calculation below. **c** A binary neural network with an input, a hidden, and an output layer. **d** An equivalent mechanical binary neural network of **c**. **e** The mechanical binary neural network in the experiment. **f** Three typical computation steps when the mechanical binary neural network is used for judging the parity of input Morse code numbers. **g** The error evolution during the training process for a BNN to distinguish labeled images of handwritten digits.

is odd (even), the output state of the memory unit in the third layer will be 0 (1). As an example, Fig. 4f gives the mathematical process to judge the parity of the input Morse code number 1.

It should be emphasized that the proposed MBNN is reprogrammable and the physical layout does not limit its function. Different working requirements can be fulfilled by simply changing the state of the memory units that store the weight and bias. We demonstrated that the MBNN in Fig. 4e is capable of other functions, for example determining whether an input Morse code number belongs to the section [4, 8] (Supplementary Movie 1).

To further illustrate the versatility of the idea underlying the MBNN, we mimicked another BNN with 2 nodes in the input layer, 2 nodes in the hidden layer, and 1 node in the output layer. Note that, in total, there are 16 types of truth table entries possible with two inputs and one output. To show the fitting ability of the corresponding MBNN, we demonstrated the realization of all these truth table entries

by reprograming the weights and biases, as shown in the Supplementary Note and Movie 2.

An MBNN with more memory units could be used for more advanced functions. In Fig. 4g, we show the training process of a BNN to distinguish between labeled images of handwritten digits (from the MNIST database[44]). The error of the training and testing sets gradually reduces and becomes 1 after about 170 training steps. Considering the equivalence of the BNN and MBNN, this result shows that the mechanical in-memory computing architecture can provide a strategy to design intelligent deformation input-output relationships that can even adapt to unseen conditions. More details of the training process are given in the Supplementary Note.

## Mechanical self-learning perceptron
The aforementioned MBNN was realized via external computing devices, but the proposed in-memory mechanical computing

architecture can enable systems that learn by themselves can also be developed. The Rosenblatt perceptron[45], which can be trained to solve linearly separable classification problems, is a classical model for supervised learning. We constructed a mechanical, self-learning perceptron inspired by Rosenblatt's strategy.

In a Rosenblatt perceptron model with one input and bias (Fig. 5a), the forward propagation procedure can be written as: $y = \varepsilon(\tilde{w}x + \tilde{b} - 1)$ with the weight and bias of backward propagation being updated by $\tilde{w} = \tilde{w} + sign(y_t - y)xd\tilde{w}$ and $\tilde{b} = \tilde{b} + sign(y_t - y)db$, where $x$, $y$, and $y_t$ are the input, output, and the target output

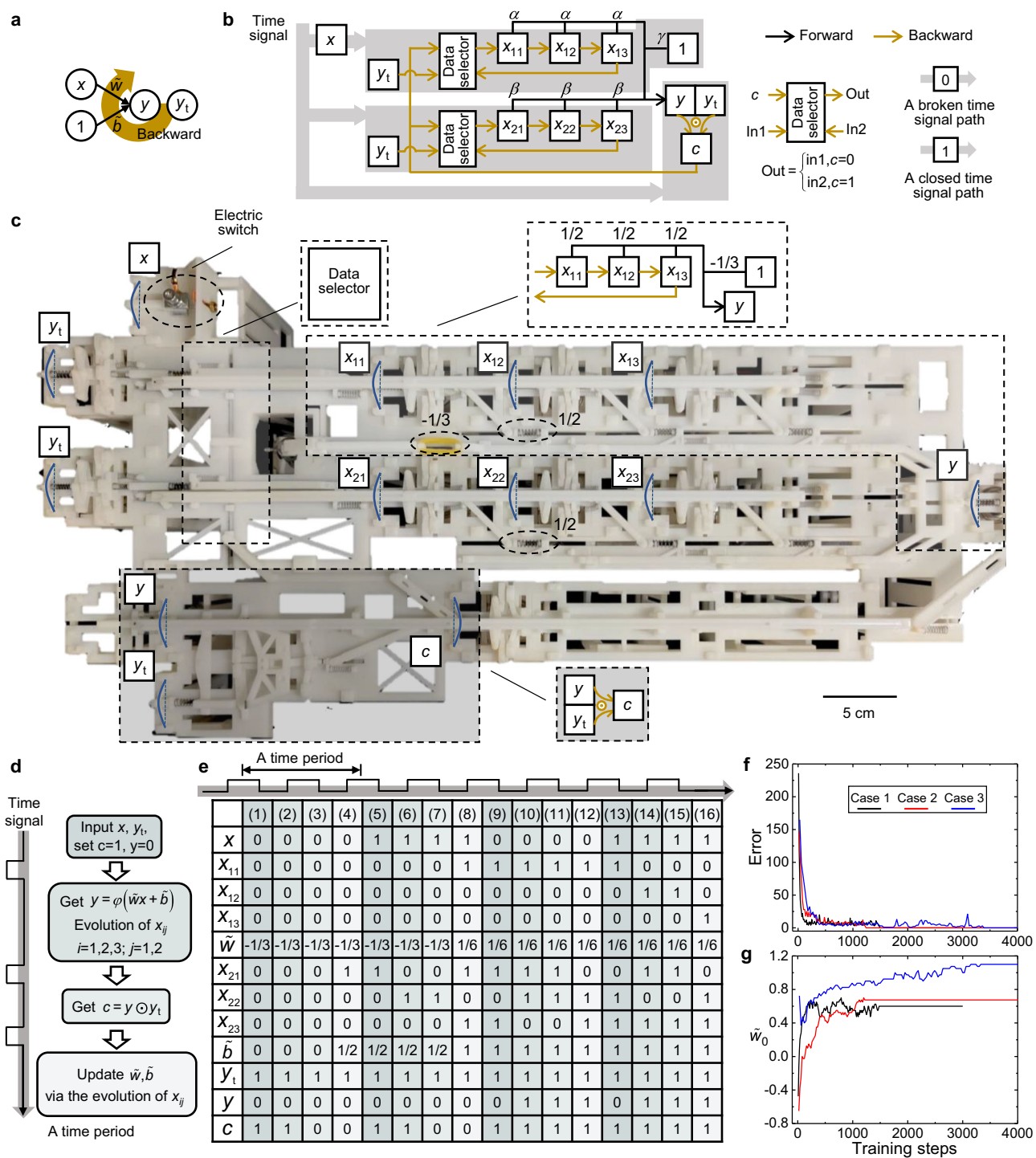

**Fig. 5 | Mechanical self-learning perceptron. a** A perceptron model with one input and bias. The weight and bias can be updated in the backward propagation process. **b** An equivalent mechanical self-learning perceptron with one input $x$ and bias. The mechanical interactions related to the backward propagation process are represented by the symbols shown in brown. The data selector outputs the input data corresponding to the control information $c$. The mechanical memory $x$ can also serve as a switch to determine whether a time signal can be transmitted to a certain part of the system. **c** A self-learning mechanical perceptron with one input and bias in the experiment. The functions of some typical parts are marked. **d** Four computation steps in a self-learning time period of the mechanical perceptron. **e** A typical evolution process of the mechanical self-learning perceptron. It can gradually acquire the target input-output relationship in a supervised learning paradigm. Here, the target input-output relationships are: input $x = 0$ output $y = 1$ and input $x = 1$ output $y = 1$. **f, g** The evolution of error and weight during the learning process for the mechanical perceptron with 10 inputs for case 1, 20 inputs for case 2, and 30 inputs for case 3.

$(x, y, y_t \in B)$, $\tilde{w}$ and $\tilde{b}$ are the weight and bias ($\tilde{w}$ and $\tilde{b}$ are real numbers), and $d\tilde{w}$ and $d\tilde{b}$ are the updating increments ($d\tilde{w}$ and $d\tilde{b}$ are greater than zero and could be different in different updating steps). Assuming that the input sets identified by different target output values are linearly separable, $y(x)$ will finally converge to $y_t(x)$ in the backward procedure[46].

The corresponding mechanical, self-learning, perceptron model has a forward propagation procedure represented in Fig. 5b (black interaction symbols). By considering the weight $\tilde{w}$ (bias $\tilde{b}$) as the interaction strength applied to the output unit $y$ by the connecting springs related to the memory units marked with $x_{11}, x_{12}, x_{13}$ and 1 ($x_{21}$, $x_{22}$, and $x_{23}$), so that $\tilde{w} = \gamma + \alpha \sum_{i=1}^{3} x_{1i}$ and $\tilde{b} = \beta \sum_{i=1}^{3} x_{2i}$, the forward procedure of the mechanical self-learning perceptron matches that of the Rosenblatt perceptron (see proof in Methods).

Furthermore, the backward propagation procedure of the mechanical self-learning perceptron is represented by the interaction symbols in brown in Fig. 5b. Once the corresponding mechanical memory units receive the time signal, they evolve according to the result $c$ of XNOR (i.e., $c = y \odot y_t$) under the control of the data selector. The evolution process is as follows:

$$\begin{cases} x_{i1} = x_{i3}, x_{i2} = x_{i1}, x_{i3} = x_{i2}, c = 1 \\ x_{i1} = y_t, x_{i2} = x_{i1}, x_{i3} = x_{i2}, c = 0 \end{cases} \quad (2)$$

During the evolution process, the weights and biases in the mechanical self-leaning perceptron update following a variant of the process used in the Rosenblatt perceptron (demonstrated in Methods), which ensures their convergence during computation and provides a theoretical basis for mechanical self-learning.

A fully functional mechanical self-learning perceptron was 3D printed and demonstrated, with $\alpha = 1/2$, $\beta = 1/2$, and $\gamma = -1/3$ (Fig. 5c, see Supplementary Note for details). The functions of some typical components are marked. All memory units were in state 0 initially in the experiment, except $c = 1$. The system learned target input-output relationships by repeatedly conducting four steps for one time period (see Fig. 5d). Note that these four steps were conducted successively by imposing the time signal on different parts of the system at different times (see Supplementary Movie S3 for the learning procedures for all possible target input-output relationships). In the first step, the input $x$ and the corresponding target output $y_t$ were entered; $c$ was set to 1. In the second step, the output $y$ was computed. Noting that $c$ was preset to 1, the memory units ($x_{11}, x_{12}, x_{13}$) and ($x_{21}, x_{22}, x_{23}$) formed two end-to-end shift register loops (Eq. (2)). Thus, the weight $\tilde{w}$ and bias $\tilde{b}$ remained unchanged, while the related memory units evolved. In the third step, an XNOR operation between the computed $y$ in the second step and the target output $y_t$ was conducted; i.e., $c = y \odot y_t$. Finally, in the fourth step, the memory units related to the weight and bias evolved following Eq. (2) and the weight and bias were updated. For the case in which the target input-output relationship was to output 1 when the input was 0 or 1, the learning process and state evolution of all memory units are listed in Fig. 5e. The weight and bias converged to 1/6 and 1, respectively, and the system achieved the desired input-output relationship.

The proposed mechanical self-learning perceptron can also be extended to cases with more inputs (see the Supplementary Note). The learning processes for three cases with 10, 20, and 30 inputs are shown in Fig. 5f and g in terms of the evolution of the error and weight $\tilde{w}_0$, respectively. After about 3500 training steps, the weight converges and the three mechanical self-learning perceptron reach their target. These examples further show the generality of the mechanical self-learning perceptron.

Biomimetic, "intelligent" functions requiring retrieval and storage, such as implementing neural networks and self-learning

behavior, have to date been realized only with electronic control and computing devices[47,48]. The proposed in-memory mechanical computing architecture enables these data-centric functions on a mechanical platform and may serve as a foundation for embedding microelectronic devices that realize even more advanced functions. The explicit mathematical model for input-output relationships in in-memory computing mechanical systems may simplify the design of such systems.

## Discussion

The in-memory mechanical computing platform enabled the integration of binary mechanical memory units and computing units, driven by time signals (external forces) and determined by interactions amongst memory units. These interactions provide a neuromorphic and function-complete method to compute mechanically within the network of memory units, analogous to in-memory computation within the human brain. The platform was demonstrated with 3D printed, in-memory computing devices, including a reprogrammable, mechanical binary neural network, and a mechanical, self-learning perceptron.

The coordination between distributed data read-write interfaces and the computing process may be beneficial for adaptive and intelligent deformation control. The absence of long-range data transfer is promising neuromorphic decision-making and biomimetic self-learning mechanical systems. Although the systems 3D printed for the demonstrations in this paper are on a scale of centimeter, the underlying physical mechanisms and strategy are scale-free. Advances in multi-material 3D printing techniques[18,49] may enable further miniaturization of such mechanical systems.

Applications of the proposed mechanical in-memory computing architecture could include robotics with neuromorphic operations in extreme environments where many electronics may not be suitable. The distributed memory units in the computing architecture coordinate with the distributed sensors and actuators of a robotic, simplifying the design of mechanical signal transmission networks and reducing the maximal data traffic in a signal path compared with centralized computing architectures[17]. The memory units themselves can also serve as sensors capable of in-situ data storage. For example, the bulked beam can be used for tactile sensing and determining whether the contacting force exceeds a certain threshold. The time signal driving the computing can be provided by the distributed actuators (corresponding to the electromagnets in this paper). The stored data is processed with the motion of the robotic. In turn, the memory units can also serve as a switch (like the one marked by $x$ in Fig. 5b, c) to control the actuators. By doing so, a storage, computing device, and movement-generating hardware interaction network is established. Considering the system works in an asynchronous mode, the results of the shallow memory layers (indicating whether an event has been triggered) can be used to control the actuators related to deeper memory layers. Thus, such robotics can be event-driven and suitable for resource-constrained scenarios. In general, the in-memory mechanical computing architecture can serve as an intelligent mechanical skeleton for embedding microelectronic devices, benefitting the construction of intelligent robotics and metamaterials[48,50–52].

## Methods

### Mechanics of the buckled beam without and with a spring

The buckled beam serving as a binary mechanical memory unit is obtained by pre-compressing a straight beam (Supplementary Fig. 2a). Its geometry parameters are listed, with $L, t, b, d$ denoting the length, thickness, width, and pre-compressing distance of the beam, respectively. The deflection as a function of position $x$ is given by $w(x)$. The ends of the beam are fixed. Its midpoint is subject to a transverse displacement load while rotation of the midpoint is prohibited. Under

this condition, the force-deflection response of the buckled beam is given by[41]

$$f(\eta) = 4\sqrt{\frac{E^2 I^2 \eta^5 (d - d_p(\eta))}{L^5(2\eta - 12\tan(\frac{\eta}{4}) + \eta\sec^2(\frac{\eta}{4}))}} \qquad (3)$$

$$w_m(\eta) = -\frac{L^3}{EI\eta^2}\left(\frac{1}{4} - \frac{1}{\eta}\tan\left(\frac{\eta}{4}\right)\right)f(\eta) \qquad (4)$$

where $\eta$ is a parameter larger than $2\pi$, $w_m(\eta)$ is the midpoint deflection of the beam, $f(\eta)$ is the reaction force, $E$ is the elasticity modulus, $I = bt^3/12$, and $d_p(\eta) = \eta^2 I/Lbt$. When $\eta$ reaches $4\pi$, the compression response will lie between that predicted by the parametric Eq. (3) and a lower reaction force given below.

$$f(w_m) = -\frac{64\pi^2 EI w_m}{L^3} \qquad (5)$$

The buckled beam is made of elastic thermoplastic polyurethanes (TPU) with a measured modulus $E$ being 72.0 MPa. The finite element method (FEM) and theoretical predicted mechanical responses are given in Supplementary Fig. 2b for the geometrical parameters given by $L = 30$ mm, $b = 2.5$ mm, $t = 2$ mm, and $d = 1$, 1.5, and 1.7 mm, showing excellent agreement. The initial deflection of the beam's midpoint is denoted as $w_0$, while the maximal value of the reaction force is represented as $f_s$. To make the binary state of the buckled beam more recognizable, $w_0$ should be larger than $L/10$. Besides, $f_s$ cannot be too large to hinder state changes of the beam. Thus, considering the driving force provided by the electromagnet in the computing process is $f_e$ (4.5 N), $f_s$ should be smaller than $f_e$. To reduce the design space of the buckled beam, $L$ and $b$ are set to be 30 mm and 2.5 mm. The contours of $w_0$ and $f_s$ in the $t$-$d$ space are given in Supplementary Fig. 2c and Supplementary Fig. 2d, respectively, with the dashed lines representing the corresponding $w_0 = L/10$ and $f_s = f_e$. With the help of these two contours, a suitable set of geometric design parameters can be selected. For the buckled beam used to construct the in-memory mechanical computing system, the set of parameters of $t = 2$ mm and $d = 1.5$ mm are adopted in the experiment, denoted by the star symbols in Supplementary Fig. 2c, d.

To design the springs used in the mechanical interaction structures, the mechanical response of the buckled beam with a connected spring is of great importance. There are two typical mechanical models. In the first model, the buckled beam is connected to fixed support via the spring of stiffness $k$ (Supplementary Fig. 2e). The corresponding mechanical response with different $k$ is shown in Supplementary Fig. 2f. It is found that if $k >$ (<) 529 N/mm, the mechanical response is mono-stable (bi-stable). For the shift register structure, the right end of spring 1 is fixed when receiving a time signal and the buckled beam should be mono-stable at this time (Fig. 2c–e). Thus, the stiffness of the spring here is selected as 588 N/mm (>529 N/mm). However, the input buckled beams of the mechanical XNOR structure should be bi-stable to store binary information though connected to a spring with a fixed end (Supplementary Fig. 1a). Thus, the stiffness of spring 1 in the XNOR structure is set as 370 N/mm (<529 N/mm).

In the second model, the buckled beam is compressed via a spring (Supplementary Fig. 2g), where $u$ represents the displacement of the left end of the spring. For several selected stiffness of the connected spring $k$, the relationship of the compressive displacement $u$ and the deflection of the buckled beam's midpoint $w_m$ is given in Supplementary Fig. 2h, where $\Delta$ is the displacement load that drives the deformation of buckled beams used for the in-memory mechanical computing system (shown in Figs. 2 and 3) and is set to be about 7 mm ($2w_0$). It can be found that if $k >$ (<) 529 N/mm, the state of the buckled

switches (does not switch) when subject to the displacement load $\Delta$. Thus, the critical stiffness $k^*$ defined in Fig. 3g is 529 N/mm. Besides, $\Delta_I$ is the displacement load that initializes the input buckled beams to state 0 for the mechanical XNOR structure (Supplementary Fig. 2b). $\Delta_I$ is set to be 4 mm (slightly larger than $w_0$). If $k >$ (<) 303 N/mm, the state of the buckled switches (does not switch) under the displacement load $\Delta + \Delta_I$. Note that the stress condition of the input buckled beams in Supplementary Fig. 1a is equivalent to that of the buckled beam in Supplementary Fig. 2g imposed by the displacement load $\Delta$. To ensure that the buckled beam can be initialized if furtherly loaded by the displacement load $\Delta_I$, the stiffness of spring 1 in the XNOR structure should be larger than 303 N/mm. Therefore, the stiffness of 370 N/mm in the design is a suitable choice.

Note that the stiffness of springs can be determined by their radius $R$, wire radius $r$, and the number of coils $N$. We measured the stiffness of springs with different geometrical parameters experimentally. The fitted stiffness (N/mm) is obtained to be:

$$k = 1.0234 \times 10^8 \frac{r^4}{4N(R - r)^3} \qquad (6)$$

With the help of this equation, we can get the springs of desired stiffness by changing the radius $R$, wire radius $r$, and the number of coils $N$.

## Estimation of the maximal clock frequency

The maximum clock frequency of the time signal shown in Fig. 1 should not be greater than the natural frequency of the buckled beam. It can therefore be estimated as follows. For a clamped-clamped straight beam, the natural frequency of the lowest order is $3.559\sqrt{EI/\rho AL^4}$ where $E$ is Young's modulus, $I$ the bending moment of inertia, $A$ the area of the cross-section of the beam, $L$ the beam length, and $\rho$ the mass density (Roark and Young, 2020. Roark's Formulas for Stress and Strain). By introducing a proportional factor, we can evaluate the natural frequency of the buckled beam as $f_b = \lambda\sqrt{EI/\rho AL^4}$, where $\lambda$ is related to the dimensionless pre-compressing distance of the beam ($d/L$) and can be obtained from FEM simulations as:

$$\lambda = -5408\left(\frac{d}{L}\right)^3 + 537.8\left(\frac{d}{L}\right)^2 + 50.93\frac{d}{L} + 3.528 \qquad (7)$$

For the buckled beam in this paper, $f_b = 1155$ Hz is obtained. In the experimental verification, the clock rate is set as 0.5 Hz, much smaller than 1155 Hz, thus ensuring all the operations can be settled. This frequency $f_b$, though much less than that of conventional electronic devices, can be substantially increased by reducing the dimensions of the beam for practical applications of intelligent matter.

## The equivalence between BNN and MBNN

To show the equivalence between BNN and MBNN, consider the number of $(\bar{x}_i^l, \bar{w}_{ij}^l)$ ($(x_i^l, w_{ij}^l)$) pairs in Fig. 4a (b) where $\bar{x}_i^l = \bar{w}_{ij}^l$ ($x_i^l = w_{ij}^l$) is $\bar{m}$ ($m$). Their forward procedure can be rewritten as:

$$\bar{x}_j^{l+1} = sign(2\bar{m} - n) \qquad (8)$$

$$x_j^{l+1} = \varepsilon(\alpha m - 1) \qquad (9)$$

When the mechanical model is equivalent to the non-mechanical one, the critical value (denoted by $m^*$) of $\bar{m}$ and $m$ for the output binary state changing should be the same. Consequently, one has:

$$\begin{cases} 2m^* - n = 0 \\ \alpha m^* - 1 = 0 \end{cases} \qquad (10)$$

Thus, $\alpha$ can be obtained as:

$$\begin{cases} \alpha = \frac{2}{n+1}, n = 2k+1 \\ \alpha = \frac{2}{n}, n = 2k \end{cases}, k = 1,2,3\dots \tag{11}$$

We set $\alpha$ (the dimensionless stiffness of the connecting springs) as (11) in the construction of the mechanical binary neuron. Then, it can be seen that when setting $x_i^l = (\bar{x}_i^l + 1)/2$ and $w_{ij}^l = (\bar{w}_{ij}^l + 1)/2$, the mechanical version of the binary neuron outputs $x_j^{l+1} = (\bar{x}_j^{l+1} + 1)/2$, which shows that the mechanical binary neuron can have an equivalent binary input-output function to the binary neuron model.

### The equivalence between the Rosenblatt perceptron and the mechanical self-learning perceptron

The weight and bias of the mechanical self-learning perceptron are defined as the interaction strength applied to the output unit as $\tilde{w} = \gamma + \alpha \sum_{i=1}^{3} x_{1i}$, $\tilde{b} = \beta \sum_{i=1}^{3} x_{2i}$. However, the input memory unit $x$ serves as a switch (Fig. 5b). Only when $x = 1$ can the time signal reach the weight memory units (see the time signal influence scope as the gray shadow), i.e., there is no interaction between the memory units storing the weight and the output unit if $x = 0$. Accordingly, the total interaction strength on the output memory unit can be represented by $\tilde{w}x + \tilde{b}$. Thus, the forward procedure of the mechanical self-learning perceptron fulfills that of the Rosenblatt perceptron.

Considering the number of the memory units storing the weights (bias) and being in state 1 is $n_w$ ($n_b$), the weight (bias) updates with the changing of $n_w$ ($n_b$) during the evolution process of Eq. (2) for the backward procedure. When $y(x) = y_t(x)$, i.e., $c = 1$, the memory units storing the weight ($x_{11}, x_{12}, x_{13}$) and bias ($x_{21}, x_{22}, x_{23}$) form two end-to-end shift register loops. Thus, $n_w$ and $n_b$ remain unchanged as also the weight and bias. When $y(x) \neq y_t(x)$, i.e., $c = 0$, $n_w$ ($n_b$) increases by $y_t - x_{13}$ ($y_t - x_{23}$) and the weight (bias) increases by $\alpha |y_t - x_{13}| sign(y_t - y)$ ($\beta |y_t - x_{23}| sign(y_t - y)$). Further considering that the memory unit $x$ controls the time signal, the updating rule of the weight and bias can be written as:

$$\begin{cases} \tilde{w} = \tilde{w} + \alpha |y_t - x_{13}| sign(y_t - y)x \\ \tilde{b} = \tilde{b} + \beta |y_t - x_{23}| sign(y_t - y) \end{cases} \tag{12}$$

Considering $d\tilde{w} = \alpha |y_t - x_{13}|$, $d\tilde{b} = \beta |y_t - x_{23}|$, Eq. (12) is a variant version of the updating process of the Rosenblatt perceptron, which ensures the convergence of weight and bias during computation and provides a theoretical basis for mechanical self-learning. More discussion about the mechanical self-learning perceptron, including the convergence analysis of the updating procedure and extension of the system to the condition with $n+1$ inputs, can be found in the Supplementary Note.

### Fabrication of the mechanical model

The mechanical memory units (buckled beams) and interaction structures are all designed in the CAD software Solidworks (Dassault Systèmes) and exported as STL files to be used in the subsequent 3D printing. The buckled beam is made of thermoplastic polyurethanes (TPU) and printed using the fused deposition modeling technique on an Ultimaker S3 printer. The springs are made of spring steel. Other components (the links, sliders, supports, and so on) are all made of photosensitive resin (DSM IMAGE8000) and printed using a Stereolithography Apparatus (SLA) UnionTech LT_450_409_G 3D printer. Then, the mechanical computing systems are obtained by assembling all the components. Below the support of each buckled beam, there is a connected electromagnet KK-1050B (Kakcom) that provides the time signal (periodic external force) under the control of a microcontroller. Besides, petroleum jelly is applied to the surface of all the components to minimize friction.

## Data availability

Source data are provided with the paper. Other findings of this study are available from the corresponding author upon request.

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

## Acknowledgements
C.Q.C. acknowledges support from the National Natural Science Foundation of China (Nos. 12132007, and 11921002) and the constructive discussion on the paper with Prof. Guy M. Genin of Washington University in St Louis.

## Author contributions
C.Q.C. designed and supervised the research. T.M. proposed the conception, carried out the structural design and experimental work. C.Q.C and T.M. wrote the manuscript and designed the figures.

## Competing interests
The authors declare no competing interests.
