## [Peer Review File · Nature Communications]

In-memory mechanical computingEditorial Note: This manuscript has been previously reviewed at another journal that is not operating a transparent peer review scheme. This document only contains reviewer comments and rebuttal letters for versions considered at *Nature Communications*.

REVIEWERS' COMMENTS

Reviewer #1 (Remarks to the Author):

Thank you for addressing my comments! I am satisfied with the revised manuscript.

Reviewer #2 (Remarks to the Author):

In reviewing the first version of the manuscript, I had two primary concerns. First, the work appeared to suffer from a lack of methodological rigor; or, at least, only the conceptual, qualitative aspects of the work were presented in the main text. Second, beyond these methodological limitations, the concepts themselves may be of limited impact. My larger concern remains the former (I still have some concern about the second point, but I defer to the editors on that).

Despite the revision, the main text is still mostly conceptual. There are far more substantive results in the SI. It remains unclear to me why the authors do not include more of it in the main text. The main text absolutely must explain what the authors have done in their study. The SI can contain details about how they performed the work, but the authors should not defer answers to the question "what did you do?" to the SI. For example, the plots of Fig. 2b,d are described in both the figure caption and in the main text as being "theoretical force-displacement" plots. Theoretical? According to what theory? Euler buckling? Something else? Why is this not stated? Moreover, in their response to reviewers, the authors contradict this by stating that "Figs. 2b and 2d have been replotted using the numerical data." Which numerical data? Finite element analysis? Surely the authors must know that numerical data is not the same as "theory". This lack of clarity leads to a superficial quality of the entire work. A well-trained reader should be able to read the main text and know what the authors have done, and what techniques they used, without having to struggle through SI and 5-minute videos. As a result, the manuscript still feels too much like a conceptual, qualitative work.

Figures 4 and 5 have been improved by including actual data. That is appreciated.

My sense is still that this would be a better fit for a journal like *Advanced Intelligent Systems*. For *Nature Comms* I believe the work should have either (1) more profound fundamental novelty or (2) a better practical demonstration of why this approach would be useful in applications.

Reviewer #3 (Remarks to the Author):

I've seen this paper before, and I have read through the authors' responses to all reviewer questions, and I am happy at the improvement in the paper. I only have the following minor comments.

'Thus, the historical data influence 95 subsequent decision-making in the mechanical system, which is a foundation of 96 intelligence.' I'd dispute this. I'd say it's the foundation of learning (specifically learning based on past events). Many things have been claimed as the foundation of intelligence (e.g. relational reasoning, or learning itself have been claimed as the foundation of intelligence and I am sure that there are more examples), so I think you need a reference or make a smaller claim.

Some symbols in the maths haven't survived the conversion to pdf, so make sure that you check this closely.

Responses to reviewers' comments

In the following, the authors' replies, summarized in a one-to-one manner, are shown in **blue**, with changes made in the revised manuscript denoted in **red**.

Referee #1

Comment of Referee #1:

Thank you for addressing my comments! I am satisfied with the revised manuscript.

Authors' reply: We would like to thank the reviewer very much for his/her invaluable time evaluating our paper.

Referee #2

Comment 1 of Referee #2:

In reviewing the first version of the manuscript, I had two primary concerns. First, the work appeared to suffer from a lack of methodological rigor; or, at least, only the conceptual, qualitative aspects of the work were presented in the main text. Second, beyond these methodological limitations, the concepts themselves may be of limited impact. My larger concern remains the former (I still have some concern about the second point, but I defer to the editors on that).

Authors' reply: The reviewer's comments are greatly appreciated. In his/her comments on the original version of our paper, the reviewer has suggested that we should move parts of the quantitative analysis in the Supplementary Information (SI) to the main text to avoid a wrong impression that our study is mainly conceptual. Accordingly, in the previous revision (referred to as Rev-1 in the following) we have moved parts of quantitative results in SI to the figures of the main text and added two more subsections in Methods of the main text, i.e., a theoretical analysis of the buckled beam with and without a spring and a theoretical estimation of the maximal clock frequency. By doing this, the section of Methods of Rev-1 consists of 5 subsections and 7 pages (pages 16–23) and enough

methodology is given to show that our study is not only conceptual but also quantitative.

For Rev-1, the reviewer still asked for more quantitative results to be included in the main text instead of in the SI. However, we would prefer not to move more results from SI to the main text. The reasons are as follows. First, in Rev-1, the most important numerical results of the paper have already been added (See, Figs. 2b, 2d, 4g, 5f, and 5g). Second, the two added subsections in Methods have provided sufficient details, including 4 pages and Eqs. (3–7), for the main text to be self-explanatory. Finally, if more quantitative results are moved from SI to the main text, our paper may become unsuitable for Nature Communications, a journal targeting general readers, and contradicts the reviewer’s Comment 4. For those who are interested in the technical details, the SI should be consulted.

As to the impact and novelty of our results, they have already been explicitly recognized by not only Referee #1 and #3 but also Referee #2 on our original paper.

Note that one of the major bottleneck problems of modern computers is data traffic. This is especially true for mechanical computing, in which signal transmission is via mechanical deformation and motion and is more difficult compared to electron transportation in silicon-based computers. Our main contribution is that an in-memory mechanical computing architecture is proposed, which enhances data exchange between mechanical computing and memory modules. The communication (interaction) between these modules is also designed to be neuromorphic and ensure a vast function design space. Then, neuromorphic functions and data-centric intelligent tasks, where wide bandwidth of data propagation is needed, can be done more easily, as demonstrated by the two examples (the MBNN and mechanical self-learning perceptron). It is expected that the architecture can facilitate the development of mechanical systems with neuromorphic intelligence, such as artificial intelligent robotics.

Comment 2 of Referee #2:

Despite the revision, the main text is still mostly conceptual. There are far more substantive results in the SI. It remains unclear to me why the authors do not include more of it in the main text. The main text absolutely must explain what the authors have done in their study. The SI can contain details about how they performed the work, but the authors should not defer answers to the question “what did you do?” to the SI. For example, the plots of Fig. 2b, d are described in both the figure caption and in the main text as being “theoretical force-displacement” plots. Theoretical? According to what

theory? Euler buckling? Something else? Why is this not stated? Moreover, in their response to reviewers, the authors contradict this by stating that “Figs. 2b and 2d have been replotted using the numerical data.” Which numerical data? Finite element analysis? Surely the authors must know that numerical data is not the same as “theory”. This lack of clarity leads to a superficial quality of the entire work. A well-trained reader should be able to read the main text and know what the authors have done, and what techniques they used, without having to struggle through SI and 5-minute videos. As a result, the manuscript still feels too much like a conceptual, qualitative work.

Authors' reply: Please see our reply to Comment 1 of Referee #1 for the reasons why we did not include more details in the main text: providing enough methodology for general readers while keeping the paper concise. We, therefore, leave most of the technical details in the SI.

The plots of Figs. 2b and 2d are based on Eqs. (3–6) given in Methods and are indeed theoretical. These equations are based on the Euler buckling theory of beams. We are sorry for confusing the reviewer in the response letter by saying “Figs. 2b and 2d have been replotted using the numerical data”, where “numerical” should read “theoretical”.

The following change is made to lines 134–135 of Page 6.

(details of the corresponding Euler buckling based mechanics are given in the Method)

Comment 3 of Referee #2:

Figures 4 and 5 have been improved by including actual data. That is appreciated.

Authors' reply: The reviewer’s suggestion of including actual data in the figures is greatly appreciated. In addition, we have also included actual data in Fig. 2b and d.

Comment 4 of Referee #2:

My sense is still that this would be a better fit for a journal like Advanced Intelligent Systems. For Nature Comms I believe the work should have either (1) more profound fundamental novelty or (2) a better practical demonstration of why this approach would be useful in applications.

Authors' reply: We believe our work has a significant impact on mechanical computing from a scientific point of view and should also be of interest to the general readers of Nature Communications.

The proposed in-memory mechanical computing architecture has addressed the bottleneck of data traffic of mechanical computing. Successful applications have been demonstrated in two data-centric neuromorphic tasks: a reprogrammable mechanical binary neural network and a mechanical self-learning perceptron. We believe that this architecture will enable the development of mechanical systems with neuromorphic intelligence, such as artificial intelligent robotics.

Please see our reply to Comment 1 of Reviewer #2 for a more detailed explanation.

Referee #3

General comment of Referee #3:

I've seen this paper before, and I have read through the authors' responses to all reviewer questions, and I am happy at the improvement in the paper. I only have the following minor comments.

Authors' reply: We are grateful to the reviewer for his/her invaluable time in evaluating our paper. The comments are of great help in improving our paper.

Comment 1 of Referee #3:

‘Thus, the historical data influence subsequent decision-making in the mechanical system, which is a foundation of intelligence.’ I’d dispute this. I’d say it’s the foundation of learning (specifically learning based on past events). Many things have been claimed as the foundation of intelligence (e.g., relational reasoning, or learning itself have been claimed as the foundation of intelligence and I am sure that there are more examples), so I think you need a reference or make a smaller claim.

Authors' reply: We agree with the reviewer. The sentence has been revised to have a softer tone as follows (lines 94–96).

Thus, the historical data influence subsequent decision-making in the mechanical system, which is a foundation of **learning**^{10,11}.

Comment 2 of Referee #3:

Some symbols in the maths haven't survived the conversion to pdf, so make sure that you check this closely.

Authors' reply: We would like to thank the reviewer for pointing out the errors in the PDF generated by the manuscript submission system. We will double-check the pdf to ensure there are no such errors this time.